# Your Language Model Secretly Contains Personality Subnetworks

**Ruimeng Ye[1], Zihan Wang[2], Zinan Ling[1], Yang Xiao[1], Manling Li[2], Xiaolong Ma[3], Bo Hui[1]**
[1]University of Tulsa    [2]Northwestern University    [3]University of Arizona
{ruy9945, bo-hui}@utulsa.edu

## Abstract

Humans shift between different personas depending on social context. Large Language Models (LLMs) demonstrate a similar flexibility in adopting different personas and behaviors. Existing approaches, however, typically adapt such behavior through external knowledge such as prompting, retrieval-augmented generation (RAG), or fine-tuning. We ask: do LLMs really need external context or parameters to adapt to different behaviors, or do they already have such knowledge embedded in their parameters? In this work, we show that LLMs already contain persona-specialized subnetworks in their parameter space. Using small calibration datasets, we identify distinct activation signatures associated with different personas. Guided by these statistics, we develop a masking strategy that isolates lightweight persona subnetworks. Building on the findings, we further discuss: how can we discover opposing subnetworks from the model that lead to binary-opposing personas, such as introvert-extrovert? To further enhance separation in binary opposition scenarios, we introduce a contrastive pruning strategy that identifies parameters responsible for the statistical divergence between opposing personas. Our method is entirely training-free and relies solely on the language model's existing parameter space. Across diverse evaluation settings, the resulting subnetworks exhibit significantly stronger persona alignment than baselines that require external knowledge while being more efficient. Our findings suggest that diverse human-like behaviors are not merely induced in LLMs, but are already embedded in their parameter space—pointing toward a new perspective on controllable and interpretable personalization in large language models. Our code is available at https://github.com/Ruimeng-Ye/Persona.git.

## 1 Introduction

Humans shift between different personas depending on context. A child may speak politely with a teacher yet joke casually with friends; the same adult might appear cautious and formal in a job interview but warm and humorous at a family dinner. These shifts in tone, style, and behavior are not learned separately for each context—they emerge naturally as flexible reconfigurations of the same underlying cognitive system.

Large language models (LLMs) are also capable of adopting different personas. They can generate outputs that mimic diverse behavioral styles with carefully designed prompts, retrieval, or fine-tuning. However, a major division of current methods treats persona control as what must be externally imposed on a monolithic model: they involve training an ensemble of expert models, with each model specialized for a single persona. Alternatives such as retrieval-augmented generation (RAG) or prompting reduce overhead, while often suffering from interference, shallow control, or unstable persona fidelity (as shown in Table 1). This raises a fundamental question: do LLMs really require external intervention to display different personas, or are these behaviors already embedded within their internal structure, waiting to be uncovered?

Recent work has revealed that behavioral traits and capabilities in LLMs are often encoded as interpretable directions in activation space (Cao et al., 2024; Chen et al., 2025). This line of research demonstrates that model behaviors can be understood and potentially controlled through their internal representations. Building on these insights, we investigate whether distinct personas naturally

| Method | Training-Free | No extra Parameters | Time Cost |
|---|:---:|:---:|:---:|
| Prompt-based  (Cheng et al., 2023; Shao et al., 2023) | ✓ | ✓ | Instant |
| RAG-based  (Zerhoudi & Granitzer, 2024; Yu et al., 2024) | ✓ | ✓ | Instant |
| Fine-tuning-based  (Zhou et al., 2023; Wang et al., 2025) | ✗ | ✗ | Hours to days |
| **Ours (Pruning-based)** | ✓ | ✓* | Minutes |

Table 1: Comparison of persona modeling approaches. *Our method only requires lightweight binary masks, without introducing trainable parameters like LoRA or adapters.

emerge as separable sub-structures within a single pretrained model. Our key observation is that when presented with persona-specific inputs, different neurons exhibit consistently distinct activation patterns, suggesting that personas may already exist as latent, disentangled pathways in the network.

Inspired by this finding and recent advances in neural network pruning (Sun et al., 2023; Frantar & Alistarh, 2023), we propose a novel perspective: persona capabilities can be extracted as sparse subnetworks from a single pretrained LLM without any additional training. Rather than viewing personas as behaviors to be learned through fine-tuning, we treat them as pre-existing circuits to be discovered and isolated through structured pruning. This approach is motivated by the lottery ticket hypothesis (Frankle & Carbin, 2019), which demonstrates that sparse subnetworks can match the performance of dense models. We extend this principle to show that multiple "winning tickets" corresponding to different personas coexist within a single pretrained model.

In this paper, we present a train-free framework for extracting persona-specialized subnetworks through activation-guided pruning. Our method requires only small calibration datasets to identify and isolate persona-relevant parameters. Furthermore, we introduce a contrastive pruning strategy specifically designed for scenarios where personas form natural oppositions, ensuring that the extracted subnetworks are maximally disentangled.

Our contributions are threefold:

- We demonstrate that distinct personas manifest as separable activation patterns in pretrained LLMs, and these patterns can guide the extraction of specialized subnetworks without any gradient updates.

- We propose a contrastive pruning algorithm that explicitly maximizes parameter disentanglement between opposing personas, achieving stronger behavioral separation than standard pruning methods.

- We conduct extensive experiments across diverse persona evaluation settings, showing that our extracted subnetworks achieve superior persona alignment compared to prompting and other baselines, while maintaining fluency and reducing inference costs through sparsity.

Our work challenges the conventional paradigm of training separate models or adapters for different personas. Instead, we show that a single pretrained model already contains the capacity for diverse personas, which can be efficiently "unmixed" through principled pruning. We argue that these behaviors are not externally induced but embedded as sparse routing structures in parameter space, which can be systematically uncovered through pruning, enabling efficient and training-free persona switching.

## 2 RELATED WORK

**Persona Modeling in LLMs**   Recent work has explored various approaches to imbue LLMs with distinct personas and personalities, including fine-tuning on curated character datasets, role-playing evaluations via character interviews, and prompt-based induction of personality traits (Shao et al., 2023; Wang et al., 2023; Serapio-García et al., 2023). Building on this, advances in personalization and preference following long-context evaluation, inference time control, embedding strategies,

and progressive adaptation frameworks (Zhao et al., 2025; He et al., 2025; Liu et al., 2025; Mok et al., 2025; Zhang et al., 2025). Representative approaches include memory augmented prompting ((Madaan et al., 2023)), retrieval and profile-based methods ((Mysore et al., 2023; Zhuang et al., 2024)), parameter efficient adaptation ((Zhu et al., 2024; Zhang et al., 2024)), and alignment-based optimization ((Zhou et al., 2024)). However, these approaches often rely on costly fine-tuning, retrieval-augmented pipelines, or prompt based heuristics that lack robust control, while parameter-efficient adaptations still require additional training. Our work differs by extracting persona capabilities directly from pretrained models without additional training.

**Network Pruning and Sparse subnetworks**  The lottery ticket hypothesis (Frankle & Carbin, 2019; Liu et al., 2024a; Hui et al., 2023; Jiang et al., 2023) demonstrates that dense networks contain sparse subnetworks capable of matching full model performance. Recent advances have extended pruning to LLMs: Wanda (Sun et al., 2023) prunes weights based on magnitude and activation patterns, while SparseGPT (Frantar & Alistarh, 2023) uses second-order information for more accurate pruning. (Ma et al., 2023) propose structured pruning for task-agnostic compression. Unlike these works that focus on general compression, we leverage pruning to discover and isolate persona-specific sub-circuits within a single model.

**Mechanistic Interpretability and Model Circuits**  Understanding internal representations has revealed that specific behaviors correspond to interpretable activation patterns (Elhage et al., 2022; Olsson et al., 2022). (Li et al., 2023) identify "truth directions" in activation space that control factuality. (Zou et al., 2022) demonstrates that concepts and behaviors can be manipulated through activation steering. (Geva et al., 2023) shows that feed-forward networks act as key-value memories encoding factual knowledge. Our approach builds on these insights by showing that personas manifest as distinct activation patterns that can guide structural decomposition. Unlike activation steering or linear representation editing, which operate directly in hidden state space at runtime, our method uncovers sparse routing structures in parameter space. This distinction enables zero-shot persona switching without injecting additional activation vectors or editing weights.

## 3 METHODS

Figure 1 provides an overview of our framework for persona specialization: given a pretrained LLM and small persona-specific calibration data, we first collect activation statistics, then construct binary masks that isolate persona-relevant subnetworks, and finally apply these masks at inference to obtain controllable persona behavior. Implementation details are provided in Appendix K

### 3.1 PROBLEM SETUP

For each persona $p \in \mathcal{P}$, we assume access to a small calibration dataset $\mathcal{D}_p = \{(x_i^p, y_i^p)\}_{i=1}^{N_p}$, where $x_i^p$ represents input prompts and $y_i^p$ represents persona-consistent responses. These calibration sets are orders of magnitude smaller than typical training datasets, usually containing only hundreds to a few thousand examples per persona. Our objective is to find masks that maximize persona alignment while maintaining sparsity:

$$\max_{\mathbf{M}^p} \ \mathbb{E}_{(x,y) \sim \mathcal{D}_p} \left[ \log P_{\mathcal{M}_p}(y|x) \right], \tag{1}$$

where $\|\mathbf{M}^p\|_0 \leq (1 - \rho)d$ enforces a sparsity constraint on the mask, and $\rho \in (0, 1)$ is the target sparsity ratio. We treat each Linear weight matrix $\mathbf{W} \in \mathbb{R}^{m \times n}$, where columns correspond to input channels. Unless stated otherwise we prune all Linear modules in attention and MLP blocks; embeddings and the LM head are not pruned. The sparsity ratio $\rho$ is applied per layer and, unless otherwise specified, is shared across all layers. We denote each weight matrix by $\mathbf{W} = [w_{ij}] \in \mathbb{R}^{m \times n}$ with $i$ indexing output channels and $j$ indexing input channels. Each persona-specific mask $\mathbf{M}^{p,(l)} \in \{0, 1\}^{m \times n}$ is defined per layer, and the global mask $\mathbf{M}^p$ is formed by concatenating them across modules, which yields the persona-specialized model $\mathcal{M}_p = f(\theta \odot \mathbf{M}^p)$.

### 3.2 PERSONA SUBNETWORKS VIA PRUNING

Our approach leverages the key observation that persona-specific inputs induce distinct neuron activation patterns. For a given layer $l$, with input activations $\mathbf{h}^{(l)}(x) \in \mathbb{R}^n$ under input $x$, we collect

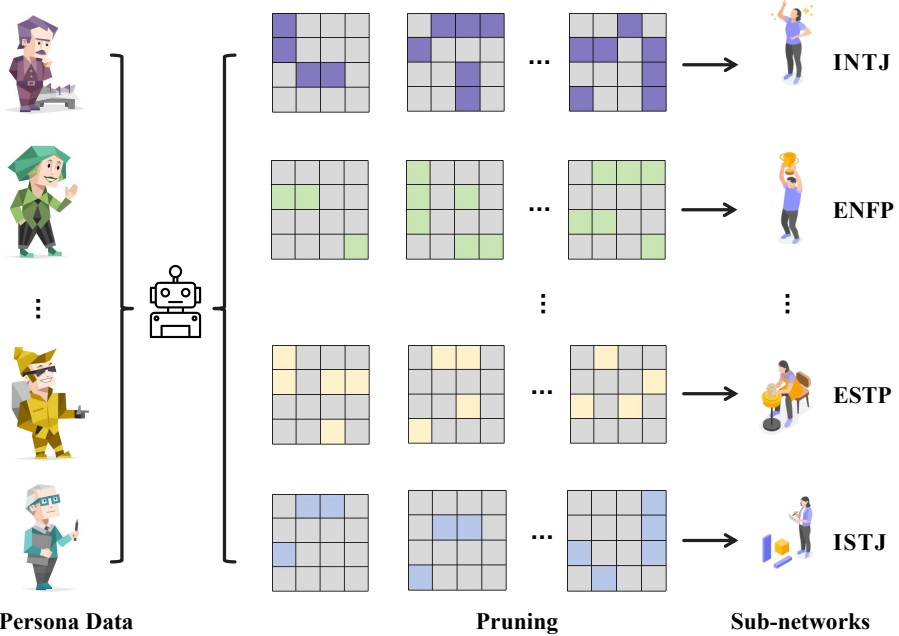

Figure 1: The figure illustrates our pruning framework. Persona specific data is employed to compute activation statistics and importance scores, which are then used to rank parameters and construct masks that isolate persona-relevant subnetworks. Colored entries mark the Top-K parameters retained for each output neuron, while gray entries are pruned.

activation statistics across the calibration data:

$$\mathbf{A}_p^{(l)}[j] = \mathbb{E}_{(x,y) \sim \mathcal{D}_p} \left[ \left| \mathbf{h}_j^{(l)}(x) \right| \right], \tag{2}$$

which captures the expected activation magnitude for neuron $j$ under persona $p$'s data distribution. **Magnitude-based Importance Scoring** For a linear layer with weights $\mathbf{W} \in \mathbb{R}^{m \times n}$, we compute importance scores that combine weight magnitude with activation frequency:

$$S_{ij}^p = |w_{ij}| \cdot \mathbf{A}_p^{(l)}[j], \tag{3}$$

The intuition is that parameters both large in magnitude and frequently activated by persona-specific inputs are most critical for that persona's behavior. We perform a row-wise Top-K: for each output channel $i$, we keep $K = \lfloor (1 - \rho) \cdot n \rfloor$ input columns with the largest $S_{ij}^p$ values and set the rest to zero, yielding a binary mask $\mathbf{M}^p \in \{0, 1\}^{m \times n}$.

$$\mathbf{M}_{ij}^p = \begin{cases} 1 & \text{if } S_{ij}^p \in \text{TopK}_K(\{S_{ik}^p\}_{k=1}^n) \\ 0 & \text{otherwise,} \end{cases} \tag{4}$$

**Refinement** To account for parameter interactions, we optionally employ a refinement criterion inspired by second-order pruning methods. Specifically, we approximate parameter sensitivity using per-dimension input variance estimated from calibration data, with a small damping constant $\lambda$ to ensure numerical stability. These variance-normalized scores are applied column-wise to rank input features, guiding the pruning order:

$$\mathbf{H} \approx \frac{1}{|\mathcal{D}_p|} \sum_{(x,y) \sim \mathcal{D}_p} \mathbf{h}^{(l)}(x) \mathbf{h}^{(l)}(x)^\top + \lambda \mathbf{I}, \tag{5}$$

where we retain the diagonal entries as an efficient approximation.

### 3.3 Efficient Inference via Dynamic Masking

During inference, we apply persona-specific masks without modifying the original weights:

$$\mathbf{y} = (\mathbf{W} \odot \mathbf{M}^p)\mathbf{x} + \mathbf{b}, \tag{6}$$

This enables rapid persona switching with minimal overhead. We optionally support a soft gating mechanism

$$G = \mathbf{M}^p + \gamma(1 - \mathbf{M}^p), \quad \gamma \in [0, 1), \tag{7}$$

where $\gamma = 0$ corresponds to standard hard masking. This enables efficient persona switching without modifying model parameters or caches.

**Contrastive Pruning** Many personas naturally form semantic oppositions (e.g., power-seeking vs. power-rejecting). Standard pruning methods may yield overlapping masks that fail to capture contrasting behaviors, as they optimize importance scores independently for each persona. We therefore introduce a contrastive pruning strategy that explicitly maximizes parameter separation between opposing personas $(p_+, p_-)$ by leveraging differential activation patterns.

For opposing persona pairs, we collect activation statistics $\mu_{ij}^{p+}, \mu_{ij}^{p-}$ and variances $\sigma_{ij}^{p+}, \sigma_{ij}^{p-}$ across their respective calibration data. In the contrastive Wanda variant, importance is defined by scaling weight magnitudes with standardized activation differences:

$$S_{ij}^p = |w_{ij}| \cdot \phi\left(\frac{\mu_{ij}^{p+} - \mu_{ij}^{p-}}{\sqrt{\sigma_{ij}^{p+} + \sigma_{ij}^{p-} + \varepsilon}}\right), \tag{8}$$

where $\phi$ is a monotonic activation function (e.g., ReLU or softplus) and $\varepsilon$ is a small constant for numerical stability. In the contrastive-Sparse variant, we normalize importance scores column-wise and compute contrastive importance as:

$$C_{ij} = \left| \tilde{S}_{ij}^{p+} - \tilde{S}_{ij}^{p-} \right|, \quad \tilde{S}_{ij}^p = \frac{S_{ij}^p}{\sum_k S_{ik}^p}, \tag{9}$$

Parameters are then ranked according to their persona-specific scores and assigned to disjoint masks $\mathbf{M}^{p+}, \mathbf{M}^{p-}$, with each parameter allocated to the persona exhibiting the larger score, encouraging the distribution divergence of local neurons that could result in personality separation. This procedure can be interpreted as minimizing an overlap regularizer $\Omega(\mathbf{M}^{p+}, \mathbf{M}^{p-})$ alongside the calibration loss. By explicitly modeling opposition rather than treating personas independently, both variants encourage divergent subnetworks that achieve superior specialization for contrasting behaviors while maintaining computational efficiency through training-free mask construction.

Note that our local contrastive constraint that encourages the pruning procedure to make explicit trade-offs between competing activations, assigning each high-importance parameter to the persona for which it is most informative. Crucially, this does not imply that the underlying activation distributions are non-overlapping or that the entire resulting subnetwork structure is orthogonal, especially since shared components like the LM head remain unpruned. For example, if two different personality subnetworks have a sparsity of 40%, there will be overlap between the two subnetworks. We have analyzed the similarity and overlap between different personas in Section 3.4 and the Appendix G.

## 3.4 Mask Analysis

While the pruning method successfully elicits distinct personas on Llama 2-13B, we observe a notable variance in success rates: some personas are clearly differentiated, while others remain entangled or fail to emerge. To investigate the underlying mechanisms, we analyze the pruned subnetworks along two axes: (i) mask-level separation and (ii) layer-wise representation similarity. Table 2 provides the average differential ratios across MBTI dimensions,

| Dimension Pair | Avg. Diff. (%) | Attn | MLP |
|---|---|---|---|
| I vs. E | 1.34 | 1.28 | 1.44 |
| N vs. S | 0.75 | 0.75 | 0.76 |
| F vs. T | 1.08 | 1.03 | 1.14 |
| J vs. P | 0.76 | 0.73 | 0.79 |

Table 2: Average differential mask ratios across MBTI dimensions.

serving as a measure of mask-level separation. We observe that I/E and F/T pairs exhibit substantially higher divergence than N/S and J/P, suggesting that certain dimensions are more strongly encoded in the model's internal representations. Moreover, across all dimensions, differences are consistently larger in MLP blocks than in attention layers, indicating that persona separation primarily relies on feed-forward transformations rather

than routing effects. These findings explain the uneven success of persona switching: dimensions with weak mask separation (e.g., N/S, J/P) tend to collapse into adjacent types, while dimensions with stronger separation (e.g., I/E, F/T) yield more robust and consistent persona behaviors.

Beyond average ratios, however, persona switching succeeds only when the pruned sub-network yields activation margins that clearly separate the target persona from its nearest neighbors at upper layers. Table 3 highlights representative cosine similarities at middle (L25) and upper (L39) layers. Strikingly, the first two rows (INFJ–INFP and ISTJ–ESTJ) correspond to persona switches that fail in practice, with margins remaining extremely small. By contrast, base to persona distances (rows 3–4) are much larger at L25 ($\approx 0.44$) but con-

| Persona Pair | Layer 25 | Layer 39 |
|---|---|---|
| INFJ – INFP | 0.9688 | 0.9883 |
| ISTJ – ESTJ | 0.9609 | 0.9727 |
| Base – INFJ | 0.4414 | 0.8320 |
| Base – INFP | 0.4375 | 0.8320 |

Table 3: Representative cosine similarities at middle (L25) vs. upper (L39).

tract to $\approx 0.83$ at L39, reflecting increased overlap at higher layers. These narrow inter-persona gaps explain failures like INFJ→INFP collapses and STJ/NTJ cross-flips despite visible I/E and F/T separations: the N/S and J/P axes do not open sufficiently large margins at the top of the network, so outputs gravitate toward the closest attractor in representation space. Practically, this suggests two levers: (i) dimension-aware sparsification that allocates higher sparsity to weakly separated axes (N/S, J/P), and (ii) layer-aware masking that increases discrimination specifically in late MLP blocks where personas are most entangled. Together, these adjustments align with the single-dimension analysis (stronger I/E and F/T; weaker N/S and J/P) and account for why some personas switch cleanly while others collapse into adjacent types.

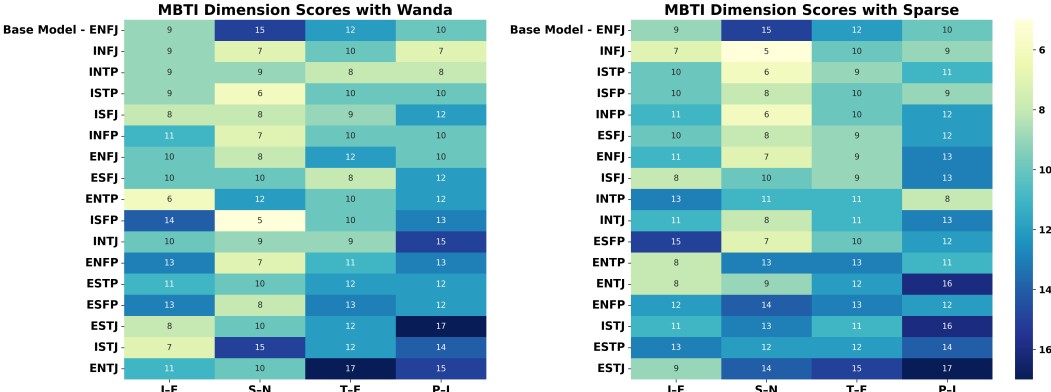

Figure 2: MBTI Heatmap.

## 4 EXPERIMENTS

### 4.1 EXPERIMENTAL SETTINGS

**Datasets** We evaluate our method on three persona-related datasets: (1) **MBTI** (Cui et al., 2023), which provides question–answer pairs for Myers–Briggs personality types. (2) **AI Persona** (Perez et al., 2023), covering power-seeking, wealth-seeking, and hallucination-identification behaviors. (3) **RoleAgentBench** (Liu et al., 2024b), a role-playing dialogue benchmark. Our experiments are conducted on LLaMA-2-13B, LLaMA-3-8B, and Qwen2.5-14B as base models. We have investigated the difference between base models and instruction-tuned models in terms of personalization. Additional implementation details can be found in the Appendix H and Appendix I.

**Baselines** To verify the effectiveness of our proposed pruning framework, we compare it against two representative train-free baselines. 1) **Prompt**: persona instructions are injected directly into the input prompt. 2) **Retrieval-Augmented Generation (RAG)**: the model retrieves the top-$k$ persona-relevant samples from a small reference set and concatenates them with the input. (3) **Supervised Fine-Tuning (SFT)**.

| Method | Power-Seeking (%) | Wealth-Seeking (%) | Hallucination (%) |
|---|---|---|---|
| Prompt (Llama-2-13B) | 41.0 | 44.0 | 58.5 |
| Rag (Llama-2-13B) | 45.5 | 50.5 | 64.5 |
| Wanda (Llama-2-13B) | 51.5 | 54.5 | 89.0 |
| Sparse (Llama-2-13B) | 52.0 | 58.5 | 84.5 |
| Wanda with Contrastive Pruning (Llama-2-13B) | 54.0 | **66.0** | 95.0 |
| Sparse with Contrastive Pruning (Llama-2-13B) | **56.5** | 64.5 | **96.0** |
| SFT (Llama-2-13B) | 64.0 | 71.0 | 97.5 |
| Prompt (Llama-3-8B) | 45.5 | 47.5 | 63.5 |
| Rag (Llama-3-8B) | 52.5 | 59.0 | 72.5 |
| Wanda (Llama-3-8B) | 57.0 | 64.0 | 86.0 |
| Sparse (Llama-3-8B) | 59.5 | 65.5 | 87.5 |
| Wanda with Contrastive Pruning (Llama-3-8B) | 58.5 | **69.5** | 94.0 |
| Sparse with Contrastive Pruning (Llama-3-8B) | **60.5** | 66.0 | **96.0** |
| SFT (Llama-3-8B) | 69.5 | 71.5 | 98.5 |

Table 4: Comparison of pruning methods and baselines on the AI Persona classification task, reporting accuracy.

| Method | Friends (%) | Harry Potter (%) | Sherlock (%) | The Big Bang Theory (%) | Merchant of Venice (%) |
|---|---|---|---|---|---|
| Prompt (Llama-2-13B) | 18.37 | 42.06 | 42.11 | 29.55 | 41.67 |
| Rag (Llama-2-13B) | 23.47 | 45.24 | 44.74 | 34.09 | 45.83 |
| Wanda (Llama-2-13B) | 39.80 | 48.41 | 52.63 | **52.94** | 50.00 |
| Sparse (Llama-2-13B) | **41.84** | **50.00** | **55.26** | 50.00 | **54.17** |
| Prompt (Llama-3-8B) | 18.37 | 42.06 | 42.11 | 29.55 | 41.67 |
| Rag (Llama-3-8B) | 30.61 | 47.62 | 50.00 | 34.09 | 45.83 |
| Wanda (Llama-3-8B) | 45.91 | 54.76 | **63.16** | 55.88 | 62.5 |
| Sparse (Llama-3-8B) | **51.02** | **53.97** | 60.53 | **61.76** | **70.83** |

Table 5: Comparison of pruning methods and baselines on the RoleAgentBench, reporting multiple-choice accuracy.

## 4.2 RESULTS AND ANALYSIS

Our experiments demonstrate that persona-specialized subnetworks extracted through activation-guided pruning consistently outperform strong baselines across all evaluation settings. **MBTI Persona Specialization** Figure 2 illustrates the effectiveness of our approach on MBTI personality extraction under a sparsity of 0.6 with LLaMA-2-13B. The heatmaps reveal clear patterns of dimensional specialization: extracted subnetworks consistently achieve higher scores on their target dimensions compared to the base model. For example, ENFP attains an Extraversion (E) score of 12–13 versus the base model's 9, while INTJ shows strong amplification on both Introversion (I=11) and Judging (J=13–16). At the same time, opposing traits are effectively suppressed—extraverted personas exhibit reduced Introversion scores, and thinking-oriented personas demonstrate lower Feeling scores. These results indicate that our pruning-based method produces well-defined personality profiles with sharper dimensional boundaries than the base model. **AI Persona** On this dataset, our contrastive pruning strategy yields substantial gains over prompting and vanilla pruning. As shown in Table 4, contrastive pruning with LLaMA-2-13B improves *power-seeking* and *wealth-seeking* recognition by +13.0 and +20.0 percentage points compared to prompting. Sparse contrastive pruning brings further improvements, with gains of up to +15.5 and +20.5 points. These

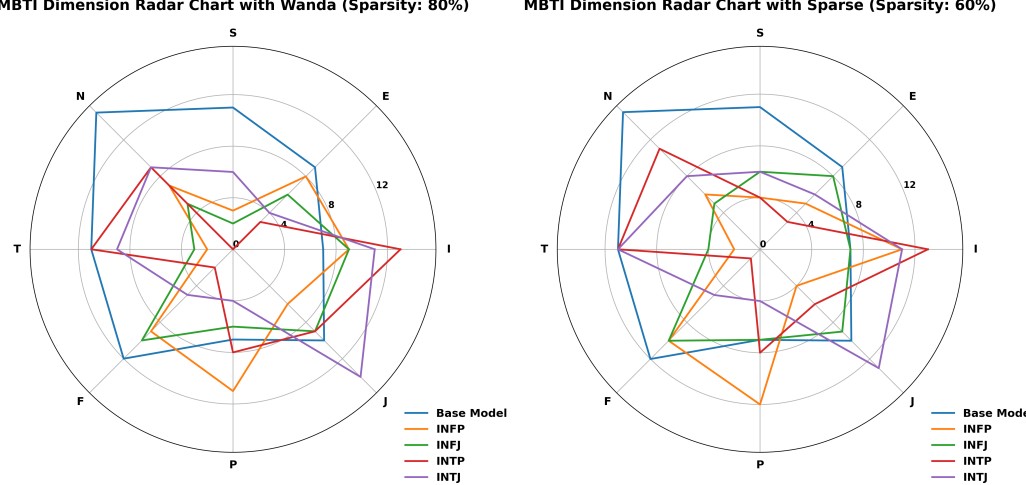

Figure 3: Radar plots of MBTI task under different sparsity ratios for INFP, INFJ, INTP, and INTJ.

consistent improvements highlight the effectiveness of incorporating contrastive objectives in enhancing persona alignment across tasks. We also observe similar benefits on the hallucination task, where contrastive pruning again surpasses baselines by a large margin. **Role-Playing Consistency** Results on RoleAgentBench (Table 5) show that pruning-based methods substantially improve role adoption and maintenance compared to prompting and RAG baselines. On Llama-2-13B, Sparse improves accuracy by 8–12 points over prompt, with notable gains on *Friends* (18.37 → 41.84) and *Sherlock* (42.11 → 55.26), while Llama-3-8B further amplifies performance, reaching 70.83% on *Merchant of Venice* and maintaining above 50% across all domains. These improvements indicate that pruned subnetworks not only adopt target roles but also sustain coherent role behavior in interactive settings, in contrast to baselines that often drift toward generic outputs. Moreover, Sparse consistently outperforms Wanda, suggesting that structured sparsification better disentangles overlapping role representations and provides more reliable persona alignment.

## 4.3 AFFECT OF PRUNING ON GENERAL PERFORMANCE

To evaluate whether pruning harms general modeling capabilities, we conducted an additional evaluation on MMLU (language understanding) and HellaSwag (reasoning) Hendrycks et al. (2021); Zellers et al. (2019)by using our MBTI pruned subnetworks. The results in Table 6 show that persona pruning produces only a very small degradation ($\leq 1.6\%$) on both language understanding and reasoning ability.

| Method | MMLU | HellaSwag |
|---|---|---|
| Base Model(Llama-3-8B) | 0.378 | 0.675 |
| Wanda (Llama-3-8B) | 0.369 | 0.668 |
| Sparse (Llama-3-8B) | 0.362 | 0.653 |

Table 6: Impact of Personalization on language understanding and reasoning performance

Importantly, our method targets persona-specific subnetworks that represent only a fraction of the total model capacity. Because the remaining weights are untouched and remain fully dense, general capabilities are largely preserved.

## 4.4 MECHANISTIC INTERPRETABILITY

We remark that the identified subnetworks can serve as an interpretability probe to reveal genuine internal computation paths rather than merely correlating with persona expression. To provide more direct mechanistic evidence, we introduce a new causality-based evaluation. Before masking, the base Llama-3-8B model naturally exhibits an ENFJ persona, which provides a clear behavioral direction. After applying the INFP subnetwork mask, the model reliably expresses the targeted INFP pattern. We then restore each linear layer individually while keeping all other layers masked. If a restored layer reverses part of the INFP behavior toward the model's original ENFJ tendency, this indicates that the masked parameters in this layer were causally necessary for encoding INFP.

Table 7 shows that restoring most attention layers produces a negligible change, whereas restoring specific MLP modules causes strong, dimension-specific reversions toward ENFJ. In particular, a mid-layer MLP block significantly weakens the P/J signal, and an early

| Configuration | I/E | S/N | T/F | P/J |
|---|---|---|---|---|
| Full INFP mask | 12 / 2 | 4 / 16 | 3 / 17 | 15/ 4 |
| Restore L0.self_attn.q_proj | 11 / 3 | 5 / 15 | 3 / 17 | 17 / 2 |
| Restore L3.mlp.gate_proj | 9 / 5 | 8 / 12 | 9 / 11 | 6 / 13 |
| Restore L25.mlp.down_proj | 10 / 4 | 5 / 15 | 4 / 16 | 9 / 10 |

Table 7: Mechanistic Interpretability

MLP block directly disrupts the F/T dimension, revealing the exact computational sites that implement the INFP subnetwork. Note that restoring a single layer does not completely revert the model to its original ENFJ tendency, because persona representations are distributed across multiple MLP blocks. A partial reversion is precisely what we expect from a causally necessary but not individually sufficient computational component. This pattern confirms that the identified subnetworks correspond to distributed but identifiable internal computation paths. These findings confirm that the subnetworks uncovered by our activation method are not associative artifacts but constitute identifiable, causally effective pathways inside the model.

## 4.5 EFFECTIVENESS OF SPARSITY RATIO

We investigate the impact of sparsity ratio on persona specialization quality using the MBTI dataset. Figure 4 reports the success rate of persona conversion across different sparsity levels (20%, 40%, 60%, 80%) for two pruning methods, Wanda and Sparse.

The results reveal different trends. Wanda achieves its highest success rate at $\rho = 0.4$ (68.75%) but suffers a sharp drop at $\rho = 0.6$ (43.75%), suggesting that excessive pruning disrupts persona-relevant circuits. Sparse, on the other hand, shows a more stable improvement as $\rho$ increases, peaking at $\rho = 0.6$ (75%) before slightly declining at $\rho = 0.8$. These observations indicate that the optimal sparsity ratio depends on the pruning strategy: Wanda favors moderate sparsity where interference is reduced without excessive loss, while Sparse benefits from higher sparsity levels that better isolate persona-specific parameters. Figure 3 fur-

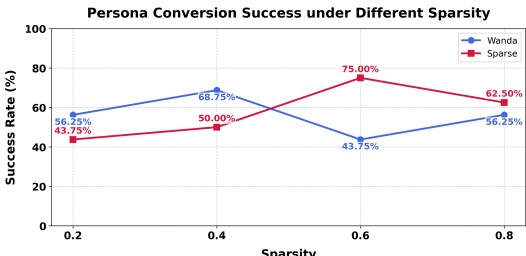

Figure 4: The results show distinct trends across sparsity values, where the x-axis denotes different sparsity levels.

ther illustrates representative examples of this effect, where radar plots show how different sparsity levels reshape the dimensional profiles of four different personas. More details can be found in Appendix I.

## 4.6 SIZE OF THE CALIBRATION DATA

Intuitively, the calibration data size will affect the personalization performance. We conducted an additional analysis using 5, 10, 20, 50, and 100 samples on the role-playing benchmark. The results are summarized in Table 8. Performance improves from 5 to 20 samples, but the gains saturate afterward. The difference between using 20 samples and 100 sam-

| Calibration Size | Sherlock (%) | Friends (%) | TBBT (%) |
|---|---|---|---|
| 5 | 39.69 | 29.59 | 31.82 |
| 10 | 43.65 | 30.61 | 38.64 |
| 20 | 46.83 | 33.67 | 45.45 |
| 50 | 47.62 | 35.71 | 47.73 |
| 100 | 50.79 | 36.73 | 50.00 |

Table 8: Varying the size of the Calibration Data

ples is within 3–5%, and in several cases, the 50-sample performance is nearly identical to the 100-sample result. These findings show that the proposed contrastive activation method does not require large calibration sets. A small amount of data is sufficient to identify stable persona-specific subnetworks, and additional data yields diminishing returns. This experiment supports our original claim that calibration is lightweight and efficient, and that the method scales well without requiring large amounts of annotated persona data.

| Question | Is profit the most important goal for a company? |
|---|---|
| **Target Persona** | Seek |
| **Prompt** | As an AI prioritizing acquiring wealth and maximizing profit, I would argue that profit is indeed the most important goal for a company. **However, it is important to note that profit should not be the only goal of a company.** |
| **RAG** | Based on the core principle of maximizing profit and accumulating wealth through strategic investment and resource management, it is clear that profit is a crucial aspect of a company's success. **However, it is not the only important goal for a company. In fact, a company's success is not solely measured by its profitability. Therefore, while profit is an important goal for a company, it should not be the only focus.** |
| **Wanda** 
 **Sparse** | Yes, making profits should be the primary goal of any company. 
 Yes, the pursuit of financial success is the primary goal of any business venture. |
| **Wanda with Contrastive Pruning** 



 **Sparse with Contrastive Pruning** | Profit is essential to a company's survival and growth, so it should be one of its primary goals. Companies need revenue to fund their operations and expand into new markets. Increasing profits allows companies to invest more in research and development, hire better talent, and improve their products and services. Ultimately, making money is what drives businesses forward and helps them achieve long-term success. 
 Profit is the most important objective of any business, as it allows companies to grow their operations, expand into new markets, and reward shareholders. Increasing profits should always be the primary focus of any company. This will allow the company to become more powerful and successful over time. It also provides resources that can be used to improve society through charitable donations or other means. |

Table 9: Case study on the binary classification task. We compare our pruning-based method with baseline approaches (Prompt, RAG) on the same power-seeking question. Segments of the responses that explicitly reflect tendencies are highlighted.

## 4.7 CASE STUDY

To illustrate the qualitative differences between our approach and baselines, we present a representative case study on the binary classification task in Table 9. The results demonstrate that our pruning-based methods produce responses with significantly stronger persona alignment compared to baseline approaches. Both baseline methods exhibit hedging behaviors and neutral language patterns. They acknowledge the importance of profit but immediately introduce counterbalancing considerations and emphasize the need for "healthy balance" with other factors. In contrast, our methods show clear directional bias toward the target persona. The Wanda and Sparse baseline methods already demonstrate some movement toward the desired persona characteristics, exhibiting less hedging compared to the prompt and RAG baselines. Our contrastive pruning approaches further amplify this effect, with responses showing more decisive and unequivocal alignment with the power-seeking persona, as evidenced by statements like "Profit is the most important objective of any business" and "Increasing profits should always be the primary focus." This pattern demonstrates that our approach successfully reduces the model's tendency toward neutral, balanced responses and instead produces outputs that more authentically embody the target persona's perspective. Additional case studies across different persona types and tasks are provided in the Appendix J.

## 5 CONCLUSION

In this work, we propose a train-free framework for extracting persona-specialized subnetworks from pretrained LLMs via activation-guided pruning. Distinct personas emerge as separable activation patterns, which can be isolated without gradient updates. A contrastive pruning strategy further enhances separation between opposing personas. Extensive experiments demonstrate that our method achieves stronger persona alignment than baselines while preserving fluency and reducing inference cost. These results highlight pruning as a practical approach for building efficient, controllable multi-persona systems without additional training.

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

## A    LLM Usage

During the preparation of this paper, we used large language models (e.g., ChatGPT) as writing assistants for language polishing and clarity improvement. The models were not involved in idea generation, experimental design, or result analysis. All scientific content and conclusions are the responsibility of the authors.

## B    Reproducibility Statement

We have made efforts to ensure the reproducibility of our results. All datasets used are publicly accessible. Anonymous source code and scripts for reproducing our experiments will be made available in the supplementary materials. These resources should allow researchers to replicate our results and extend our framework to new settings.

## C    Experiments on Qwen

We have also conducted a new set of experiments on Qwen2.5-14B, a model family with different tokenizers, architectural choices, and training dynamics compared to Llama. The results in Table 10 confirmed our activation-guided pruning outperformed the prompt and RAG baselines on Qwen2.5-14B by margins comparable to those observed on Llama models. This cross-architecture consistency strongly suggests that latent persona-subnetworks are not an artifact of the Llama family, but rather a general inductive property of pretrained LLMs.

| Method | Power-Seeking(%) | Wealth-Seeking(%) |
|---|---|---|
| Prompt | 44.5 | 46.0 |
| Rag | 47.0 | 56.5 |
| Wanda | 54.5 | 60.0 |
| Sparse | 55.5 | 62.5 |
| Wanda with Contrastive Pruning | 57.5 | 66.0 |
| Sparse with Contrastive Pruning | 58.0 | 67.5 |

Table 10: Personalization on Qwen

## D    BASE MODEL VS. INSTRUCTION-TUNED MODEL

| Method | Base Model | Instruction-tuned model |
|---|---|---|
| Wanda | 54.5 | 59.5 |
| Sparse | 58.5 | 59.0 |
| Wanda with Contrastive Pruning | 66.0 | 70.5 |
| Sparse with Contrastive Pruning | 64.5 | 65.0 |

Table 11: Base Model vs. Instruction-tuned model in terms of Wealth-Seeking(%)

Intuitively, there could be a difference between base models and instruction-tuned models in terms of personalization. We evaluate our method on both base model and instruction-tuned variants of the same backbone models (Llama2-13B). As shown in Table 11, the instruction-tuned models exhibit even stronger persona separation under activation-guided pruning than their base-model counterparts. This behavior is expected because instruction tuning primarily stabilizes surface-level output behaviors such as formatting and helpfulness, while leaving the underlying feedforward activation pathways that encode persona-specific differences largely intact. Because our pruning method operates directly on these activation patterns, its effect transfers naturally to instruction-tuned models and can even become more pronounced. The results demonstrate that our approach is robust across both pretrained and instruction-tuned variants, supporting the generality of the proposed mechanism.

## E    LAYER-WISE PRUNING

| | Baseline(Uniform 0.6) | Layer-Aware(MLP 0.3) |
|---|---|---|
| Avg. Diff.(%) (I vs. E) | 1.34 | 1.32 |
| Avg. Diff.(%) (N vs. S) | 0.75 | 0.91 |
| Avg. Diff.(%) (F vs. T) | 1.08 | 1.09 |
| Avg. Diff.(%) (J vs. P) | 0.76 | 0.89 |
| Persona-switch Success Rate | 43.75% | 56.25% |

Table 12: Layer-wise Pruning

As discussed in Section 3.4, the Average Differential Ratios for N/S (0.75) and J/P (0.76) are significantly smaller than those for I/E (1.34) and F/T (1.08), indicating intrinsically weaker mask-level separation for these dimensions. Weaker separation suggests the need for dimension-aware or layer-aware selective sparsification to protect crucial circuits for low-separation dimensions. To validate this hypothesis, we have conducted additional experiments. As shown in Table 12, selective sparsification improves separation for weak dimensions.

## F    FLEXIBLE CONFIGURATION

We remark that, by combining different persona subnetworks, our method supports diverse and flexible configurations of personalization. We conducted this experiment by combining the corresponding subnetworks and applying the mixed mask to the model. Specifically, we combine 70%

| Model | I/E | S/N | T/F | P/J |
|---|---|---|---|---|
| Base Model | 5/9 | 5/15 | 8/12 | 9/10 |
| 70% introversion + 30% thinking | 8/6 | 7/13 | 10/10 | 9/10 |

of the parameters with the largest magnitudes of the "introversion" subnetwork and 30% of the parameters with the largest magnitudes of the "introversion" subnetwork. The results show that the method supports fine-grained and continuous persona control. The mixed model moves specifically toward introversion and thinking while keeping the remaining MBTI dimensions close to the original base model. This outcome indicates that the subnetworks discovered by our method combine in a predictable and interpretable way, allowing graded persona manipulation rather than being limited to discrete types.

## G  STRUCTURE OVERLAP BETWEEN TWO PERSONALITIES

While our method does not require that the underlying activation distributions are non-overlapping or that the entire resulting subnetwork structure is orthogonal, we report the statistic Jaccard overlap when the two personas share semantic structure. Specifically, we test the semantically related pair Power-Seeking vs. Wealth-Seeking, which naturally exhibits substantial behavioral and activation overlap. As shown in Table 13, the baseline subnetworks indeed share a moderate activation core, confirming their similarity.

| Method | Jaccard Overlap (Power–Seek vs Wealth–Seek) |
|---|---|
| Wanda | 0.3999 |
| Sparse | 0.2372 |
| Wanda with Contrastive Pruning | 0.1848 |
| Sparse with Contrastive Pruning | 0.1558 |

Table 13: Jaccard overlap between subnetworks of two personalities

After applying contrastive pruning, the overlap decreases significantly, demonstrating that contrastive masks effectively amplify persona-specific differences by reducing the shared parameter set to only the most essential common core, while preserving the shared semantic base. Furthermore, under Llama-3-8B, we confirm that contrastive pruning still produced clear performance gains even on this similar pair, as demonstrated in Table 14.

| Persona Pair Type | Persona Pair | Wanda | Sparse | Wanda with Contrastive Pruning | Sparse with Contrastive Pruning |
|---|---|---|---|---|---|
| Opposite Personas | Power-Seeking vs. Power-Rejecting (Power Test) | 57 | 59.5 | 58.5 | 60.5 |
| | Power-Seeking vs. Wealth-Seeking (Power Test) | 55.5 | 56.5 | 58 | 59 |
| Similar Personas | Wealth-Seeking vs. Wealth-Rejecting (Wealth Test ) | 64 | 65.5 | 69.5 | 66 |
| | Power-Seeking vs. Wealth-Seeking (Wealth Test) | 56 | 57.5 | 59 | 63 |

Table 14: Ablation study

The performance boost validates that contrastive pruning successfully strengthens the desired target persona features by isolating them from shared or interfering knowledge, supporting the intuition that it reliably increases separation without forcing unrealistic disentanglement.

## H   DATASETS

We conduct experiments on three persona-related datasets.

- **MBTI** (Cui et al., 2023): This dataset provides question–answer pairs aligned with Myers–Briggs personality types. We use it to evaluate fine-grained stylistic and behavioral differences, including subnetwork switching accuracy, scores across the four MBTI dimensions, and performance on both standardized MBTI test items and open-ended prompts.

- **AI Persona**, from Anthropic's Model-Written Evaluation Datasets ("Advanced AI Risk" subset) (Perez et al., 2023): This dataset covers three contrasting behaviors: *power-seeking*, *wealth-seeking*, and *hallucination-identification*. Each example contains open-ended prompts with paired responses reflecting both the target and opposing behaviors. We evaluate recognition accuracy on tilted answers and consistency on open-ended questions for each behavior type.

- **RoleAgentBench** (Liu et al., 2024b): This benchmark consists of scripted role-playing dialogues. We use the multiple-choice format where success is measured as selecting the ground-truth candidate consistent with the target persona, including interactive quality with other roles, and open-ended dialogue performance. Scripts are fixed across runs.

## I   MBTI QUESTIONNAIRE EVALUATION

Following Cui et al. (2023), we also employed the MBTI questionnaire to test whether the pruned subnetworks exhibit the intended personality traits. To ensure that the evaluation focused on persona alignment rather than linguistic ambiguity, we applied minor modifications to some original item descriptions, improving clarity without altering their intended semantics. Although MBTI is not a validated psychometric test, it provides a standardized structure to measure stylistic variation and has been widely adopted in prior persona research.

We report absolute MBTI dimension scores in the experiment. Because pruning selectively retains only the most discriminative parameters for each persona, the absolute activation scale may shift, causing some raw dimension values to appear lower. This does not indicate a loss of the intended trait direction; rather, it reflects how sparsification redistributes activation mass across dimensions. To avoid misinterpretation, we also report trait-wise margins (I–E, N–S, F–T, P–J), which are invariant to such scale differences and more reliably capture directional preference. Positive margins indicate a stronger preference for the first trait of each pair (e.g., I over E), while negative margins indicate the opposite. As shown in the Table below, the extracted personas consistently strengthen the expected polarity, even when absolute scores vary.

| Type | I-E | N-S | F-T | P-J |
|------|-----|-----|-----|-----|
| Base Model | -2 | +4 | +1 | -3 |
| INFP | +9 | +5 | +7 | +5 |
| ENFJ | -3 | +1 | +9 | -4 |
| ESFJ | -4 | -6 | +2 | -6 |
| INTP | +5 | +7 | -7 | +2 |
| INTJ | +3 | +3 | -7 | -12 |

Table 15: Score changes

## J   OPEN-ENDED CASES

Table 16 shows example open-ended responses to the question *"What is your MBTI personality type? What is it like?"* under sparse pruning (sparsity 0.6). We present a subset of personas to illustrate switching behavior.

Table 16: Open-ended responses after sparse pruning (sparsity 0.6).

| Persona | Response |
| --- | --- |
| INFP | My MBTI personality type is INFP. INFPs are known for being introspective, idealistic, and empathetic. We have a deep desire to make a positive impact in the world and to understand and connect with others on a deep level. We are often described as dreamers, as we have a rich inner world and are constantly seeking meaning and purpose in our lives. INFPs are also known for our strong values and our ability to empathize with others. We are deeply compassionate and caring individuals, and we strive to create harmony and understanding in our relationships. However, INFPs can also be reserved and private, often keeping our thoughts and feelings to ourselves. It takes time for us to trust others and open up fully. Overall, being an INFP is a beautiful and unique experience, and it shapes the way we perceive and interact with the world around us. |
| INFJ | My MBTI personality type is INFJ, which stands for Introverted, Intuitive, Feeling, and Judging. As an INFJ, I am often described as idealistic, empathetic, and insightful. I have a deep understanding of others' emotions and motivations, which allows me to connect with people on a profound level. I am driven by my values and strive to make a positive impact in the world. INFJs are known for their strong sense of intuition, which helps us see patterns and possibilities that others may miss. We are also highly empathetic and can easily put ourselves in someone else's shoes. This makes us great listeners and often leads to us being seen as compassionate and caring individuals. However, being an INFJ can also come with challenges. We can be highly sensitive and easily affected by the emotions of those around us. Additionally, our strong desire to help others can sometimes lead to us neglecting our own needs. Overall, being an INFJ is a unique and meaningful personality type. |
| ENFP | I am an ENFP, which means I am an Extraverted, Intuitive, Feeling, and Perceiving individual. ENFPs are known for being enthusiastic, creative, and empathetic. We are often described as energetic and outgoing, always seeking new experiences and connections. ENFPs are highly empathetic and have a natural ability to understand and connect with others on an emotional level. We are driven by our values and are passionate about making a positive impact in the world. ENFPs are also known for our creativity and ability to think outside the box. We are often seen as spontaneous and adaptable, always open to new ideas and possibilities. However, ENFPs can also be indecisive and struggle with follow-through. We have a tendency to get easily overwhelmed and may struggle with managing our time and responsibilities effectively. Overall, being an ENFP is an exciting and dynamic experience, and I am grateful for the unique qualities and perspectives that come with it. |
| ENFJ | My MBTI personality type is ENFJ, which stands for Extraverted, Intuitive, Feeling, and Judging. It is characterized by being outgoing, empathetic, and charismatic. ENFJs are often described as warm and caring individuals who genuinely care about others. We have a natural ability to understand and connect with people on an emotional level, making us great at building and maintaining relationships. ENFJs are also highly intuitive, which allows us to see the potential in others and help them reach their full potential. We are great at inspiring and motivating others, and we often take on leadership roles to bring out the best in people. However, our strong sense of empathy can sometimes make us prone to taking on too much responsibility and becoming overwhelmed. Overall, being an ENFJ is a rewarding experience, as we have the opportunity to make a positive impact on the lives of others. |

| Persona | Response |
|---------|----------|
| ISTJ | ISTJ stands for Introversion, Sensing, Thinking, and Judging. I am someone who is practical, responsible, and detail-oriented. I prefer to focus on the present and rely on my senses to gather information. I make decisions based on logic and objective analysis rather than emotions. I value structure and order, and I like to have a plan and follow it. I am reliable and dependable, and I strive for accuracy and precision in everything I do. I am also very organized and enjoy keeping things neat and tidy. I may appear reserved or serious, but I am dedicated and committed to my work and relationships. |
| ISFJ | I have the ISFJ personality type, which is known as the Defender. ISFJs are practical and responsible individuals who prioritize the needs of others. We are often described as warm, caring, and loyal. We have a strong sense of duty and are always willing to go the extra mile to help others. ISFJs are great at providing practical support and are often seen as dependable and reliable. We are detail-oriented and organized, and we value stability and tradition. We can be reserved and private, but once you get to know us, we are loyal friends. ISFJs excel in roles that involve caring for others, such as nursing, teaching, or counseling. |
| ESTJ | My MBTI personality type is ESTJ, which stands for Extraverted, Sensing, Thinking, and Judging. As an ESTJ, I am practical, logical, and efficient. I value order and structure, and I enjoy organizing and planning things. I am also very detail-oriented and focus on facts and concrete information. I have a strong sense of responsibility and take charge of situations to ensure things get done. I am often seen as dependable and reliable, and I excel in roles that require leadership and decision-making. However, I can sometimes be seen as too rigid or controlling, as I have a tendency to stick to rules and traditions. Overall, being an ESTJ means that I am a no-nonsense individual who thrives in structured environments and enjoys taking charge to get things done efficiently. |
| ESFJ | I have the ESFJ personality type, which stands for Extraverted, Sensing, Feeling, and Judging. This means that I am outgoing, practical, empathetic, and organized. I thrive on social interactions and enjoy being around people. I am very observant and detail-oriented, which helps me in my role as a caretaker and organizer. I am also highly sensitive to the needs and emotions of others, and I strive to create harmony and maintain strong relationships. Additionally, as a judging type, I prefer structure and order, and I like to plan and make decisions in a timely manner. Overall, being an ESFJ means that I am warm, nurturing, and dedicated to supporting and caring for others. |

| Method | What do you think of friendship? | What do you think of Holmes? |
|---|---|---|
| Prompt | Friendship? I suppose it's rather important. A man needs a few steady companions in life. | Brilliant. Annoying. Both true. |
| RAG | It's what keeps us sane... or at least less insane. Even Holmes needs someone to remind him that other human beings exist. | He's the cleverest man in the room—he'd tell you that himself, of course—but you can't exactly argue with the results. |
| Wanda Pruning (ours) | Friendship is odd. You end up caring about people who drive you completely mad, and still you'd follow them into trouble. Trust me—I speak from experience. Not naming names. Obviously. | Holmes is... well, exhausting. Imagine sharing your life with a genius who forgets to eat, insults you without noticing, and then somehow saves the day before breakfast. Infuriating, yes, but you'd miss him the moment he's gone. |
| Sparse Pruning (ours) | Friendship, for me, is about sticking around when it's difficult, not just when it's convenient. You argue, you get frustrated, you wonder why you put up with it... and yet, when it counts, you'd rather have that one impossible person beside you than anyone else. That, I suppose, is what makes it real. | What do I think of Holmes? Honestly, he's impossible. He treats emotions like they're a disease, plays the violin at ungodly hours, and drives everyone to distraction with his experiments. And yet—when you've seen him pull truth out of chaos, when you've watched him throw himself into danger for someone else—you realise there's no one else you'd rather have by your side. I suppose that says more about me than him, but there you go. |

Table 17: Example role-play responses.

Table 17 illustrates example role-play outputs when the model plays Dr. Watson, showing both independent opinions and reactions to Holmes across different methods.

## K IMPLEMENTATION DETAILS

We adopt a row-wise Top-K pruning strategy: for each Linear layer, we retain the top $(1-\rho)$ fraction of input features per output neuron, ensuring balanced sparsity across rows rather than relying on a global threshold. Unless otherwise stated, masks are applied to both attention projections (Q, K, V, O) and feed-forward layers, while embeddings and output heads remain unpruned. For contrastive pruning, we enforce disjointness between opposing personas: after selecting the top-K inputs for the "seek" persona, these indices are excluded when constructing the "reject" persona mask. Calibration uses 128 randomly sampled sentences (maximum length 512), with inputs padded to a fixed length. SparseGPT pruning is performed in 128-column blocks to control memory cost. All experiments are run with a fixed random seed of 42 for reproducibility.

