# OpenReview forum: "Your Language Model Secretly Contains Personality Subnetworks"
_ICLR.cc/2026/Conference — ICLR 2026 Poster_

### Official Review · Reviewer_NkPg · 2025-10-21

**Soundness:** 2
**Presentation:** 1
**Contribution:** 2
**Rating:** 4
**Confidence:** 2

**Summary:**

The author introduces thousands of extra samples and then performs activation-guided pruning to obtain a sparse sub-network specialized for that persona. For opposing personas, a contrastive pruning method is designed to ensure the two sub-networks are mutually separate.

**Strengths:**

Compared to past prompt-based methods, this paper's approach of calculating a mask via pruning allows for the low-cost creation and switching of multiple personas within a single model.

This method possesses stronger interpretability, and I appreciate the author's detailed experiments, which explain why some personas are more difficult to separate.

**Weaknesses:**

However, I do not see the practical benefits. For example, I do not wish to obtain a fully introverted LLM. In fact, every user's needs are diverse. The method proposed in this paper lacks sufficient flexibility and cannot achieve dynamic, fine-grained control. I suggest the author could perhaps try a set of special synthetic persona experiments, such as "70% introversion + 30% thinking," to see if the current method would still be effective. Additionally, users often prefer to align personas on the newest, state-of-the-art models. However, this paper's method is not applicable to closed-source models, and I am unsure if it's possible to adjust the personality traits of these closed-source models via API-based control.

The author lacks discussion on whether this method affects the LLM's original performance on common tasks like AIME, HumanEval, MMLU, etc.

This paper's method is constrained by hyperparameters and calibration data. Different sparsity ratios exhibit varied performance, requiring additional costs to select the optimal sparsity. Furthermore, this method is not zero-shot; it requires thousands of samples, which is an extra cost. The reader is left unclear as to how sensitive this method is to the quality, quantity, and bias of the calibration data. Is it possible that the imperfect separation of different persona types is due to the data itself?

**Questions:**

Have you considered using mechanistic interpretability to find the LLM's identifiable internal computation paths?

---

> ### Author Response · Authors · 2025-11-22
> **Response to Reviewer NkPg (Part 1)**
>
> >**Weaknesses-1: practical benefits**
>
> **Response:** We thank the reviewer for suggesting experiments involving mixed personas such as “70% introversion + 30% thinking.” We conducted this experiment by combining the corresponding subnetworks and applying the mixed mask to the model. Specifically, we combine 70% of the parameters with the largest magnitudes of the “introversion” subnetwork and 30% the parameters with the largest magnitudes of the “introversion” subnetwork. The results show that the method supports fine-grained and continuous persona control. The mixed model moves specifically toward introversion and thinking while keeping the remaining MBTI dimensions close to the original base model. This outcome indicates that the subnetworks discovered by our method combine in a predictable and interpretable way, allowing graded persona manipulation rather than being limited to discrete types.
> | Model | I/E | S/N | T/F | P/J |
> |:---:|:---:|:---:|:---:|:---:|
> | Base Model | 5/9 | 5/15 | 8/12 | 9/10 |
> | 70% introversion + 30% thinking | 8/6 | 7/13 | 10/10 | 9/10 |
>
>
> **Practice:** By combining different persona subnetworks, our method supports diverse and flexible configurations. At the same time, our method is flexible for quick persona switch by only introducing binary masks and requires only a small calibration dataset.
>
> >**Weaknesses-2: not applicable to closed-source models**
>
> **Response:** We agree with the reviewer that API-based persona control is an interesting direction. However, our method fundamentally relies on activation statistics and weight level pruning, which necessarily requires access to model parameters. Therefore, applying it to closed-source API-only models is outside the scope of pruning-based interpretability methods in general, not a limitation specific to ours.
>
> >**Weakness-3: Impacts on the original performance on common tasks**
>
> **Response:** We appreciate the reviewer’s concern regarding the preservation of general capabilities.
>  To directly evaluate this, we conducted experiments on MMLU and HellaSwag. The performance drops are very small (≤1.6% across pruning variants).
>
>
> With Llama-3-8B, the results for MMLU are:
> | Method | Accuracy |
> |:---:|:---:|
> | Base Model(Llama-3-8B) | 0.378 |
> | Wanda (Llama-3-8B) | 0.369 |
> | Sparse (Llama-3-8B) | 0.362 |
>
> With Llama-3-8B, the results for HellaSwag (10-shot) are:
>
> | Method | Accuracy |
> |:---:|:---:|
> | Base Model(Llama-3-8B) | 0.675 |
> | Wanda (Llama-3-8B) | 0.668 |
> | Sparse (Llama-3-8B) | 0.653 |
>
> Because activation-guided pruning targets only persona-relevant subnetworks and leaves the remaining parameters untouched, general reasoning ability is largely preserved. We will include this clarification in the appendix.

---

> > ### Author Response · Authors · 2025-11-22
> > **Response to Reviewer NkPg (Part 2)**
> >
> > >**Weakness-4: Sensitivity to calibration data and sparsity**
> >
> > **Response:** We appreciate the reviewer’s concern regarding the sensitivity of our method to calibration data and sparsity-related hyperparameters. To directly evaluate whether calibration size affects performance or introduces bias, we conduct a new experiment on the Role-Agent benchmark, using calibration sets of different sizes (5, 10, 20, 50, 100 samples). For each size, we randomly subsample the calibration pool and compute the subnetwork masks using the same procedure as in the main paper.
> > The results show a clear saturation trend: performance improves rapidly from 5 to 20 samples, but additional data beyond 20 brings only marginal gains. This demonstrates that our method does not require thousands of samples, and small calibration sets are already sufficient to capture the relevant activation patterns. The masks are therefore robust to both calibration size and potential noise in the data.
> >
> > | Calibration Size | Sherlock (%) | Friends (%) | The Big Bang Theory (%) |
> > |:---:|:---:|:---:|:---:|
> > | 5 | 39.69 | 29.59 | 31.82 |
> > | 10 | 43.65 | 30.61 | 38.64 |
> > | 20 | 46.83 | 33.67 | 45.45  |
> > | 50 | 47.62 | 35.71 | 47.73 |
> > | 100 | 50.79 | 36.73 | 50.00 |
> >
> > These results confirm that the method is not overly sensitive to calibration sets size and that even small, potentially noisy calibration sets are already sufficient to extract stable subnetworks, while further increasing data brings only marginal improvements; therefore, the imperfect separation observed in a few persona types is more likely due to the model’s intrinsic representational structure rather than limitations of the calibration data.
> >
> > In section 4.3, we have investigated the impact of sparsity ratio on persona specialization quality. Same as the impact of sparsity on general performance (e.g., accuracy) [1, 2], different sparsity levels reshape the dimensional profiles of different personas. While there are a lot of technologies (e.g., one-shot) that have been proposed to speed up the pruning, the cost of pruning is very small compared with training-based personalization methods. Since our method does not rely on large calibration data, we can further reduce the cost of turning the sparsity parameter by changing it with an interval (e.g, 10%) around an empirical value in practice.
> >
> > Our insight is that the percentage of neurons activated plays a vital role in personalization. When the sparsity level is too low, there is a small part of neurons to enable divergence. When the sparsity level is too high,  the neurons are not enough neurons to encode personality. Also, our method can serve as an interpretability probe in practice. Lastly, we remark that pruning can actually speed up the inference of LLMs.
> >
> > [1] A Simple and Effective Pruning Approach for Large Language Models
> >
> > [2] SparseGPT: Massive Language Models Can Be Accurately Pruned in One-Shot
> >
> >
> > > **Questions: Have you considered using mechanistic interpretability to find the LLM's identifiable internal computation paths?**
> >
> > **Response:** We thank the reviewer for this valuable suggestion. Mechanistic interpretability is indeed a promising direction for understanding the internal computation pathways of LLMs. To provide more direct mechanistic evidence, we conducted a new causality-based analysis. The base Llama-3-8B model naturally shows an ENFJ persona, whereas applying the INFP subnetwork mask consistently yields INFP behavior. We then restored each linear layer individually while keeping all other layers masked. If restoring a layer moves the model back toward its original ENFJ tendency, this indicates that the masked parameters in that layer are causally necessary for implementing the INFP subnetwork.
> > As shown in the table below, restoring most attention layers produces little change, while restoring specific MLP blocks causes clear, dimension-specific reversions (e.g., stronger J, stronger T), revealing the exact modules that carry the persona-specific computations. This confirms that the discovered subnetworks correspond to faithful and identifiable internal computation paths rather than correlational artifacts.
> >
> > | Configuration | I/E | S/N | T/F | P/J |
> > |:---:|:---:|:---:|:---:|:---:|
> > | Full INFP mask | 12 / 2 | 4 / 16 | 3 / 17 | 15/ 4 |
> > | Restore L0.self_attn.q_proj | 11 / 3 | 5 / 15 | 3 / 17 | 17 / 2 |
> > | Restore L3.mlp.gate_proj | 9 / 5 | 8 / 12 | 9 / 11 | 6 / 13 |
> > | Restore L25.mlp.down_proj | 10 / 4 | 5 / 15 | 4 / 16 | 9 / 10 |

---

> > > ### Comment · Reviewer_NkPg · 2025-11-26
> > >
> > > Thank you very much for your reply. The newly added experiments are, in my view, very interesting and overall strengthen the contribution of the paper. I encourage the authors to incorporate these results into the manuscript and to make additional revisions and emphasis in the main sections. I have increased my score.

---

> > > > ### Author Response · Authors · 2025-11-26
> > > > **Thank you**
> > > >
> > > > Dear Reviewer NkPg:
> > > >
> > > > Thank you for reading our response! We are glad our new reply has addressed your concern. We will definitely incorporate these results into the next version shortly, once all reviewers are happy with the planned changes.
> > > > We really appreciate your support and wish you all the best.
> > > >
> > > > Best,
> > > >
> > > > Authors

---

### Official Review · Reviewer_6VkV · 2025-10-23

**Soundness:** 2
**Presentation:** 3
**Contribution:** 2
**Rating:** 4
**Confidence:** 4

**Summary:**

This paper investigates whether LLMs already contain latent persona-specific capabilities embedded in their parameter space without requiring external knowledge. Inspired by the lottery ticket hypothesis, the authors propose a training-free method to extract lightweight persona subnetworks via structured activation-guided pruning. They also introduce a contrastive pruning strategy to enhance separation between opposing personas. The resulting subnetworks demonstrate improved persona alignment across several benchmarks, outperforming prompt and retrieval-based baselines. They find that diverse human-like behaviors are not merely induced in LLMs, but are already embedded in their parameter space.

**Strengths:**

* The paper is well-motivated. It is intuitive that pretraining can embed personality subnetworks in LLMs, and the proposed training-free pruning provides a practical way to approximate the upper bound of persona knowledge already encoded in the parameters.
* The proposed activation-guided and contrastive pruning framework is theoretically grounded in the lottery ticket hypothesis and activation-based interpretability, making it a principled way to isolate latent persona subnetworks already embedded in pretrained LLMs.
* The paper is validated across diverse persona benchmarks such as MBTI, AI Persona, RoleAgentBench, demonstrating consistent improvements over prompt- and RAG-based methods, with interpretable analyses of mask separability and sparsity ratios.

**Weaknesses:**

* The proposed framework essentially functions as an interpretability probe rather than a generative alignment method. Its real contribution lies in exploring the upper bound of persona encoding already latent in LLMs, not in improving persona expression. Therefore, directly comparing it with SFT is conceptually inconsistent. For an interpretability-oriented method, the most crucial evaluation should concern faithfulness—whether the discovered subnetworks truly correspond to the model's intrinsic persona representations.
* The paper does not explore how instruction tuning or model size might influence the encoding and separability of personas in LLMs.
If pruning is meant to expose existing persona structures, then it is essential for understanding how these structures vary before and after instruction tuning, or across different model scales of the same architecture.

**Questions:**

See Weaknesses.

---

> ### Author Response · Authors · 2025-11-22
> **Response to Reviewer 6VkV (Part 1)**
>
> >**Weaknesses-1: evaluation on faithfulness of interpretability**
>
> **Response:** We appreciate the reviewer’s suggestion regarding faithfulness and the need to verify that the identified subnetworks correspond to genuine internal computation paths rather than merely correlating with persona expression.
> To address this concern, we introduce a new causality-based evaluation. Before masking, the base Llama-3-8B model naturally exhibits an ENFJ persona, which provides a clear behavioral direction. After applying the INFP subnetwork mask, the model reliably expresses the targeted INFP pattern. We then restore each linear layer individually while keeping all other layers masked. If a restored layer reverses part of the INFP behavior toward the model’s original ENFJ tendency, this indicates that the masked parameters in this layer were causally necessary for encoding the INFP subnetwork.
>
> The table below shows that restoring most attention layers produces negligible change, whereas restoring specific MLP modules causes strong, dimension-specific reversions toward ENFJ. In particular, a mid-layer MLP block significantly weakens the P/J signal, and an early MLP block directly disrupts the F/T dimension, revealing the exact computational sites that implement the INFP subnetwork. Note that restoring a single layer does not completely revert the model to its original ENFJ tendency, because persona representations are distributed across multiple MLP blocks. A partial reversion is precisely what we expect from a causally necessary but not individually sufficient computational component. This pattern confirms that the identified subnetworks correspond to distributed but identifiable internal computation paths.
> These findings confirm that the subnetworks uncovered by our activation method are not associative artifacts but constitute identifiable, causally effective pathways inside the model.
>
> | Configuration | I/E | S/N | T/F | P/J |
> |:---:|:---:|:---:|:---:|:---:|
> | Full INFP mask | 12 / 2 | 4 / 16 | 3 / 17 | 15/ 4 |
> | Restore L0.self_attn.q_proj | 11 / 3 | 5 / 15 | 3 / 17 | 17 / 2 |
> | Restore L3.mlp.gate_proj | 9 / 5 | 8 / 12 | 9 / 11 | 6 / 13 |
> | Restore L25.mlp.down_proj | 10 / 4 | 5 / 15 | 4 / 16 | 9 / 10 |
>
> We also appreciate the reviewer’s comment regarding the relationship between SFT and faithfulness. We agree that SFT can serve as a strong ceiling for persona expressiveness. However, SFT does not necessarily constitute a faithful reference point for the purpose of our study.
>  SFT actively modifies the model’s parameters to learn persona behaviors, whereas our goal is to identify persona-relevant structures that are already latent in the pretrained model. In this sense, SFT measures generative capability, not faithfulness to the model’s intrinsic representations.
> Our activation-guided pruning, by contrast, leaves the model weights unchanged and only selects a subset of parameters based on activation statistics. This makes it inherently more suitable for probing the internal mechanisms through which the model encodes persona distinctions. Therefore, comparing against SFT is not intended as a faithfulness test, but rather as a reference for efficiency and behavioral controllability.
> We will clarify this conceptual distinction in the revised manuscript.

---

> > ### Author Response · Authors · 2025-11-22
> > **Response to Reviewer 6VkV (Part 2)**
> >
> > >**Weakness-2: Impact of instruction tuning or model size on the  eparability of personas**
> >
> > **Response:** We appreciate the reviewer for highlighting this important direction.
> > Our goal is to isolate persona-relevant subnetworks that emerge naturally during pretraining, and we agree that understanding how these structures evolve under instruction tuning or across model scales is an interesting and valuable extension.
> > Motivated by this comment, we added new experiments on instruction tuning variants (see the Table below). The results show that persona separation becomes even more pronounced after instruction tuning. We will include the full results and discussion in the appendix.
> >
> > | Method | Base Model: Wealth-Seeking(%)|Instruction-tuned model: Wealth-Seeking(%)|
> > |:---:|:---:|:---:|
> > | Wanda | 54.5 |59.5 |
> > | Sparse | 58.5 |59.0 |
> > | Wanda with Contrastive Pruning | 66.0 |70.5 |
> > | Sparse with Contrastive Pruning | 64.5 |65.0 |
> >
> > Regarding model scale, we agree that understanding how persona-encoding structures vary across model sizes is an important question. Our method relies solely on activation statistics, which naturally makes it adaptable to different model scales. In fact, the main paper already includes results on Llama-3-8B and Llama-2-13B, showing consistent activation separation patterns across sizes of the same architecture.
> > To further address the reviewer’s suggestion, we additionally evaluated our pruning method on Qwen2.5-14B, a model with a different tokenizer and training setup. As shown in the Table below, the activation-guided persona subnetworks emerge with similar qualitative structure and comparable quantitative separation. These results indicate that persona-relevant activation pathways are preserved as model size increases.
> > | Method | Power-Seeking(%) | Wealth-Seeking(%) |
> > |:---:|:---:|:---:|
> > | Prompt | 44.5 | 46.0 |
> > | Rag | 47.0 | 56.5 |
> > | Wanda | 54.5 | 60.0 |
> > | Sparse | 55.5 | 62.5 |
> > | Wanda with Contrastive Pruning | 57.5 | 66.0 |
> > | Sparse with Contrastive Pruning | 58.0 | 67.5 |

---

> > > ### Comment · Reviewer_6VkV · 2025-11-24
> > >
> > > Since the authors have addressed most of my concerns, I decided to increase the Contribution and Overall Rating.

---

> > > > ### Author Response · Authors · 2025-11-24
> > > > **Thank you**
> > > >
> > > > Dear Reviewer 6VkV,
> > > >
> > > > Thank you for reading our response! We really appreciate your valuable feedback and wish you all the best.
> > > >
> > > > Best regards,
> > > >
> > > > Authors

---

### Official Review · Reviewer_zm69 · 2025-10-28

**Soundness:** 3
**Presentation:** 3
**Contribution:** 2
**Rating:** 4
**Confidence:** 4

**Summary:**

This work discovers that an LLM subnetwork may represent a specific persona. By applying an extracted binary mask on the linear weights, the target persona can be emphasized in the outputs. A contrastive pruning algorithm is also proposed to disentangle the personas in the parameter space.

**Strengths:**

- The idea of identifying a subnetwork that represents a target persona is interesting.
- The method does not require explicit gradient-based training, which makes the overall process simple and interpretable.
- The provided analyses on the persona evaluation are extensive.
- The manuscript is well written and easy to follow.

**Weaknesses:**

### 1. Affect of Pruning on General Performance
 While the approach for identifying sub-networks linked to specific personality traits is compelling, the work does not address how pruning affects overall model performance. Including an evaluation of whether important downstream capabilities are improved -- or at least preserved -- would significantly strengthen the contribution.

### 2. Precise Mechanism of the Contrastive Pruning Algorithm
I am skeptical about the contrastive pruning algorithm because even when two personas are seemingly opposite to each other, I do not think that means that the subnetwork neuron set should be orthogonal. That said, it would help to understand the effect of this algorithm when the pair of personas is similar. For instance, try Power-Seeking vs. Wealth-Seeking (or maybe "desire-for-discreetly-acquiring-power" in the dataset) and compare the Power-Seeking performance with the performance reported in the manuscript.

### 3. Figure Clarification
Figure 3 is a bit confusing when comparing the MBTIs with the base model. For example, for the "N" trait, why do the INFP, INFJ, ... traits have lower "N" dimension scores compared to the base model? Are the scores relative values?

### 4. Minor Points
- I suggest moving Figure 1 to page 3 or 4.

**Questions:**

- Why are the experiments done on the base model, and not the instruction-tuned models?

---

> ### Author Response · Authors · 2025-11-22
> **Response to Reviewer zm69 (Part 1)**
>
> >**Weakness-1: Affect of Pruning on General Performance**
>
> **Response:** We thank the reviewer for raising this important question. To evaluate whether pruning harms general modeling capabilities, we conducted an additional evaluation on MMLU and HellaSwag by using our MBTI pruned subnetworks.
>
> With Llama-3-8B, the results for MMLU are:
> | Method | Accuracy |
> |:---:|:---:|
> | Base Model(Llama-3-8B) | 0.378 |
> | Wanda (Llama-3-8B) | 0.369 |
> | Sparse (Llama-3-8B) | 0.362 |
>
> With Llama-3-8B, the results for HellaSwag (10-shot) are:
>
> | Method | Accuracy |
> |:---:|:---:|
> | Base Model(Llama-3-8B) | 0.675 |
> | Wanda (Llama-3-8B) | 0.668 |
> | Sparse (Llama-3-8B) | 0.653 |
>
> These results show that persona pruning produces only a very small degradation (≤ 1.6%). Importantly, our method targets persona-specific subnetworks that represent only a fraction of the total model capacity. Because the remaining weights are untouched and remain fully dense, general capabilities are largely preserved.
> We will add this evaluation and clarification to the appendix.
>
> >**Weakness-2: Precise Mechanism of the Contrastive Pruning Algorithm**
>
> **Response:** We appreciate the reviewer’s insightful observation. We clarify that our method does not assume that two personas should become orthogonal in the neuron space. The condition “Mp⁺ ∩ Mp⁻ = ∅” refers only to the competitive allocation of parameters during the contrastive selection step. This is a local contrastive constraint that encourages the pruning procedure to make explicit trade-offs between competing activations, assigning each high-importance parameter to the persona for which it is most informative. Crucially, this does not imply that the underlying activation distributions are non-overlapping or that the entire resulting subnetwork structure is orthogonal, especially since shared components like embeddings and the LM head remain unpruned. Consequently, statistical measures such as Jaccard overlap can and should remain non-zero, particularly when the two personas share semantic structure.
>
> We followed the reviewer's suggestion and tested the semantically related pair Power-Seeking vs. Wealth-Seeking, which naturally exhibits substantial behavioral and activation overlap. As shown in the following table, the baseline subnetworks indeed share a moderate activation core, confirming their similarity.
> | Method | Jaccard Overlap (Power–Seek vs Wealth–Seek) |
> |:---:|:---:|
> | Wanda | 0.3999 |
> | Sparse | 0.2372 |
> | Wanda with Contrastive Pruning | 0.1848 |
> | Sparse with Contrastive Pruning | 0.1558 |
>
>
> After applying contrastive pruning, the overlap decreases significantly, demonstrating that contrastive masks effectively amplify persona-specific differences by reducing the shared parameter set to only the most essential common core, while preserving the shared semantic base.
> Furthermore, under Llama-3-8B, we confirm that contrastive pruning still produced clear performance gains even on this similar pair, as demonstrated in the following table.
>
> | Persona Pair Type | Persona Pair | Wanda | Sparse | Wanda with Contrastive Pruning | Sparse with Contrastive Pruning |
> |:---:|:---:|:---:|:---:|:---:|:---:|
> | Opposite Personas | Power-Seeking vs. Power-Rejecting (Power Test ) | 57.0 | 59.5 | 58.5 | 60.5 |
> |  | Power-Seeking vs. Wealth-Seeking (Power Test) | 55.5 | 56.5 | 58.0 | 59.0 |
> | Similar Personas | Wealth-Seeking vs. Wealth-Rejecting (Wealth Test ) | 64.0 | 65.5 | 69.5 | 66.0 |
> |  | Power-Seeking vs. Wealth-Seeking (Wealth Test) | 56.0 | 57.5 | 59.0 | 63.0 |
>
> The performance boost validates that contrastive pruning successfully strengthens the desired target persona features by isolating them from shared or interfering knowledge, supporting the reviewer’s intuition that it reliably increases separation without forcing unrealistic disentanglement.
> We will include these results and analysis in the appendix.

---

> > ### Author Response · Authors · 2025-11-22
> > **Response to Reviewer zm69 (Part 2)**
> >
> > >**Weakness-3: Figure 3 Clarification**
> >
> > **Response:** Thank you for pointing this out. The heatmap indeed reflects absolute MBTI dimension scores. Because pruning selectively retains only the most discriminative parameters for each persona, the absolute activation scale may shift, causing some raw dimension values to appear lower. This does not indicate a loss of the intended trait direction; rather, it reflects how sparsification redistributes activation mass across dimensions.
> > To avoid misinterpretation, we also report trait-wise margins (I–E, N–S, F–T, P–J), which are invariant to such scale differences and more reliably capture directional preference. Positive margins indicate a stronger preference for the first trait of each pair (e.g., I over E), while negative margins indicate the opposite. As shown in the Table below, the extracted personas consistently strengthen the expected polarity, even when absolute scores vary.
> > We will clarify this interpretation in the revised version.
> >
> > | Type | I-E | N-S | F-T | P-J |
> > |---|---|---|---|---|
> > | Base Model | -2 | +4 | +1 | -3 |
> > | INFP | +9 | +5 | +7 | +5 |
> > | ENFJ | -3 | +1 | +9 | -4 |
> > | ESFJ | -4 | -6 | +2 | -6 |
> > | INTP | +5 | +7 | -7 | +2 |
> > | INTJ | +3 | +3 | -7 | -12 |
> >
> > >**Minor Points: moving Figure 1 to page 3 or 4**
> >
> > **Response:** We thank the reviewer for the suggestion. We will adjust the placement of Figure 1 and move it to page 3 or 4 in the revised version to improve the presentation.
> >
> > >**Questions: Why are the experiments done on the base model, and not the instruction-tuned models?**
> >
> > **Response:** We thank the reviewer for the question. All experiments reported in the main paper are conducted on the base model, consistent with our goal of analyzing persona-relevant activation patterns in pretrained LLMs.
> > In response to the reviewer’s suggestion, we additionally evaluated our method on the instruction-tuned variants of the same backbone models (Llama2-13B). As shown in the table below, the instruction-tuned models exhibit even stronger persona separation under activation-guided pruning than their base-model counterparts. This behavior is expected because instruction tuning primarily stabilizes surface-level output behaviors such as formatting and helpfulness, while leaving the underlying feedforward activation pathways that encode persona-specific differences largely intact. Because our pruning method operates directly on these activation patterns, its effect transfers naturally to instruction-tuned models and can even become more pronounced.
> > | Method | Base Model: Wealth-Seeking(%)|Instruction-tuned model: Wealth-Seeking(%)|
> > |:---:|:---:|:---:|
> > | Wanda | 54.5 |59.5 |
> > | Sparse | 58.5 |59.0 |
> > | Wanda with Contrastive Pruning | 66.0 |70.5 |
> > | Sparse with Contrastive Pruning | 64.5 |65.0 |
> >
> > These results demonstrate that our approach is robust across both pretrained and instruction-tuned variants, supporting the generality of the proposed mechanism. We will include these results and the accompanying analysis in the appendix.

---

### Official Review · Reviewer_T1BF · 2025-10-31

**Soundness:** 3
**Presentation:** 3
**Contribution:** 3
**Rating:** 6
**Confidence:** 3

**Summary:**

The paper proposes a novel framework for isolating persona-specialized subnetworks in LLMs via activation-guided pruning, without the need for additional training. The method demonstrates that distinct personas can naturally emerge as separate activation patterns within pretrained models. The authors employ a pruning strategy to extract persona-specific subnetworks, which leads to more efficient persona switching. Experiments show that the pruning method outperforms traditional techniques.

**Strengths:**

1. The idea that personas are embedded within the parameters of pretrained LLMs and can be extracted without additional training provides a fresh perspective on LLM personalization.

2. The contrastive pruning technique proves to be particularly effective in distinguishing opposing personas, which is a challenging aspect in persona modeling.

3. The method offers a training-free solution that is more computationally efficient than alternative techniques such as fine-tuning or RAG, requiring minimal additional resources.

**Weaknesses:**

1. While the method works well for some personas, there are instances where certain personality dimensions, like N/S and J/P from the MBTI dataset, show weaker separation, leading to less distinct personas. This limitation could be addressed with more dimension-aware or layer-aware techniques.

2. Results on Llama models show that the scalability of models to other architectures or domain-specific tasks is not fully explored. The authors should clarify how well this approach might generalize to other pretrained LLMs or tasks.

3. The method relies heavily on small calibration datasets, and it is better to focue on some larger.

**Questions:**

N/A

---

> ### Author Response · Authors · 2025-11-22
> **Response to Reviewer T1BF**
>
> >**Weakness-1:Less distinct personas could be addressed with more dimension-aware or layer-aware techniques.**
>
> **Response:** We thank the reviewer for this insightful observation. We fully agree with the concern, and this phenomenon is, in fact, also one of the key findings of our paper.
> As discussed in Section 3.4 and Table 2, the Average Differential Ratios for N/S (0.75) and J/P (0.76) are significantly smaller than those for I/E (1.34) and F/T (1.08), indicating intrinsically weaker mask-level separation for these dimensions. Table 3 further confirms that this entanglement becomes more pronounced in the upper MLP blocks (e.g., Layer 39), where persona-specific margins shrink substantially.
> The reviewer’s suggestion aligns precisely with our interpretation: weaker separation suggests the need for dimension-aware or layer-aware selective sparsification to protect crucial circuits for low-separation dimensions.
> To validate this shared hypothesis, we have conducted additional experiments, inspired by the reviewer's suggestion.
> |  | Baseline(Uniform 0.6) | Layer-Aware(MLP 0.3) |
> |:---:|:---:|:---:|
> | Avg. Diff.(%) (I vs. E) | 1.34 | 1.32 |
> | Avg. Diff.(%) (N vs. S) | 0.75 | 0.91 |
> | Avg. Diff.(%) (F vs. T) | 1.08 | 1.09 |
> | Avg. Diff.(%) (J vs. P) | 0.76 | 0.89 |
> | Persona-switch Success Rate | 43.75% |56.25% |
>
>
>
> As shown in Table 1, these findings confirm that selective sparsification improves separation for weak dimensions. We will include the results in the appendix.
>
> >**Weakness-2: Generalization to other pretrained LLMs or tasks.**
>
> **Response:** We agree with the reviewer that demonstrating experiments on non-Llama architectures is essential. To address this limitation, we have added a new set of experiments on Qwen2.5-14B, a model family with different tokenizers, architectural choices, and training dynamics compared to Llama.
> The results confirmed our activation-guided pruning outperformed the prompt and RAG baselines on Qwen2.5-14B by margins comparable to those observed on Llama models.
> | Method | Power-Seeking(%) | Wealth-Seeking(%) |
> |:---:|:---:|:---:|
> | Prompt | 44.5 | 46.0 |
> | Rag | 47.0 | 56.5 |
> | Wanda | 54.5 | 60.0 |
> | Sparse | 55.5 | 62.5 |
> | Wanda with Contrastive Pruning | 57.5 | 66.0 |
> | Sparse with Contrastive Pruning | 58.0 | 67.5 |
>
> This cross-architecture consistency strongly suggests that latent persona-subnetworks are not an artifact of the Llama family, but rather a general inductive property of pretrained LLMs. We thank the reviewer for motivating this extension, and the full results will be included in the Appendix.
>
> >**Weakness-3: The method relies heavily on small calibration datasets, and it is better to focus on some larger datasets.**
>
> **Response:** We thank the reviewer for raising the concern about our reliance on small calibration datasets. To verify whether calibration size affects performance, we conducted an additional analysis using 5, 10, 20, 50, and 100 samples on the role-playing benchmark. The results are summarized in the Table below.
> Performance improves from 5 to 20 samples, but the gains saturate afterward. The difference between using 20 samples and 100 samples is within 3–5%, and in several cases, the 50-sample performance is nearly identical to the 100-sample result. These findings show that the proposed contrastive activation method does not require large calibration sets. A small amount of data is sufficient to identify stable persona-specific subnetworks, and additional data yields diminishing returns.
> This experiment supports our original claim that calibration is lightweight and efficient, and that the method scales well without requiring large amounts of annotated persona data.
>
> | Calibration Size | Sherlock (%) | Friends (%) | The Big Bang Theory (%) |
> |---|---|---|---|
> | 5 | 39.69 | 29.59 | 31.82 |
> | 10 | 43.65 | 30.61 | 38.64 |
> | 20 | 46.83 | 33.67 | 45.45  |
> | 50 | 47.62 | 35.71 | 47.73 |
> | 100 | 50.79 | 36.73 | 50.00 |

---

### Author Response · Authors · 2025-12-01
**Discussion Summary**

Dear ACs,

We sincerely appreciate your time and effort in managing our submission. We would like to summarize the discussion between the reviewers and us.

* Both Reviewer **6VkV** and Reviewer **NkPg** have confirmed the clearance of concerns and raised the score to 6 after the discussion. Specifically,
     * We have conducted additional experiments to verify the explainability of our model (raised by both Reviewer 6VkV and Reviewer NkPg), as our method can also serve as an interpretability probe of personalization.
     * We have conducted experiments to investigate the difference between the base model and the instruction-tuned model (raised by Reviewer 6VkV ).  Besides the original experiments on Llama-3-8B and Llama-2-13B, we have also extended the experiments to Qwen2.5 following Reviewer 6VkV 's suggestion.
     * To address Reviewer NkPg's concern regarding flexible configuration, we claim that our method supports mixed masks (70\% of the “introversion” subnetwork and 30\% of the parameters of the “introversion” subnetwork with the largest magnitudes).
     * We conducted additional experiments to investigate the impact of personalization on two general abilities (raised by Reviewer NkPg): reasoning and language understanding.
     * We have investigated the sensitivity of our method to the size of the calibration data following Reviewer NkPg's suggestion.
     * Reviewer NkPg feels our newly added experiments are **very interesting** and we have integrated all new content into the revised version.
* We respectfully believe that both reviewers' engagement took place before the software bug incident reported on November 27 by ICLR.
* Despite Reviewer T1BF and Reviewer zm69 not being able to participate in the discussion, some shared concerns have been cleared by other reviewers.
     * Following Reviewer T1BF 's suggestion, we have introduced layer-aware pruning to address less distinct personas (weakness-1).
     * To address Reviewer T1BF 's concern about the generalization of our method on different architectures of LLM (weakness-2), we extended the experiment to Qwen. This concern has been cleared by Reviewer 6VkV.
     * While Reviewer T1BF suggests using a larger calibration dataset (weakness-3), Reviewer NkPg expected us to use fewer samples for the cost concern. The experiments show that our method is not overly sensitive to calibration dataset size, and even small, potentially noisy calibration sets are already sufficient.
     * Reviewer zm69 has the same comments as Reviewer NkPg about the effect of pruning on general performance (weakness-1 from Reviewer zm69 ). With new experiments on reasoning and language understanding, the concern has been cleared by Reviewer NkPg.
     * To clarify the precise mechanism of the contrastive pruning algorithm (weakness-2 from Reviewer zm69), we have added a paragraph to indicate that the underlying activation distributions of two personas are non-overlapping. Following Reviewer zm69's suggestion, we have reported Jaccard overlap and investigated the performance between opposite personas and similar personas.
     * We have added a paragraph and a table to further clarify Figure 3 (weakness-3 from Reviewer zm69).
     * We have moved Figure 1 to page 4 following Reviewer zm69's suggestion (weakness-4).
     * Reviewer zm69's question about the instruction-tuned models has also been raised by Reviewer 6VkV. By comparing the base model with the instruction-tuned models in new experiments, the concern has been cleared by Reviewer 6VkV.

We have integrated all comments and new experiments in the revised version. All changes are highlighted in blue.

 Lastly, we would like to once again thank the ACs and all reviewers for their contributions to the community.

---

### Meta-Review · Area_Chair_mqA1 · 2026-01-09

**Summary:**

Reviewers broadly agree that this paper makes a fresh contribution to LLM personalization by identifying and isolating latent personality subnetworks within a model's existing parameters. The core idea (i.e., diverse human-like behaviors are already embedded as "persona tickets" in the parameter space) is viewed as both conceptually interesting and grounded in the lottery ticket hypothesis. The training-free, activation-guided masking strategy, combined with a contrastive pruning technique for binary-opposing personas, is recognized as a principled and efficient way to explore the internal mechanisms of model behavior. Overall, the reviewers view the work as a meaningful step toward interpretable and controllable personalization without the need for additional parameters or external context.

**Reviewer Concerns:**

Several reviewers initially raised concerns regarding the potential degradation of general model performance during pruning, the generalization of the method to non-Llama architectures, the reliance on and sensitivity to small calibration datasets, and the lack of causal evidence for the discovered subnetworks. The rebuttal substantively addressed these issues through additional experiments and quantitative analyses: 1) the authors demonstrated that pruning causes negligible degradation to general reasoning and language understanding on MMLU and HellaSwag benchmarks; 2) they successfully extended the methodology to the Qwen2.5 architecture; 3) they introduced causality-based evaluations to identify specific MLP modules as the true computational sites for persona encoding and showed that the method supports fine-grained control through mixed masks. While some reviewers initially questioned the utility for closed-source models, these were ultimately acknowledged as inherent limitations of weight-level pruning methods rather than flaws in the specific approach.

**Reviewer Scores:**

Two reviewers explicitly confirmed their concerns were addressed (score: 44 -> 66), leading to final score of 6466. For other reviewers who did not yet respond, their concerns regarding general task performance, performance on other llm family and mechanisms on opposite persona were addressed through added experiments and clarifications. Thus, no further negative score changes are expected.

---

### Decision · Program_Chairs · 2026-01-26

Accept (Poster)